# Mining the Genome of *Bacillus velezensis* VB7 (CP047587) for MAMP Genes and Non-Ribosomal Peptide Synthetase Gene Clusters Conferring Antiviral and Antifungal Activity

**DOI:** 10.3390/microorganisms9122511

**Published:** 2021-12-03

**Authors:** Saravanan R, S Nakkeeran, N Saranya, C Senthilraja, P Renukadevi, A.S. Krishnamoorthy, Hesham Ali El Enshasy, Hala El-Adawi, V.G. Malathi, Saleh H. Salmen, M. J. Ansari, Naeem Khan, R. Z. Sayyed

**Affiliations:** 1Department of Plant Pathology, Tamil Nadu Agricultural University, Coimbatore 641003, India; saravanankavi010@gmail.com (S.R.); csenthilraja1991@gmail.com (C.S.); renucbe88@gmail.com (P.R.); milkmush@rediffmail.com (A.S.K.); vgmalathi@rediffmail.com (V.G.M.); 2Department of Plant Biotechnology and Bioinformatics, Tamil Nadu Agricultural University, Coimbatore 641003, India; saranya.n@tnau.ac.in; 3Institute of Bioproduct Development (IBD), Universiti Teknologi Malaysia (UTM), Skudai, Johor Bahru 81310, Malaysia; henshasy@ibd.utm.my; 4Faculty of Engineering School of Chemical and Energy Engineering, Universiti Teknologi Malaysia (UTM), Skudai, Johor Bahru 81310, Malaysia; 5City of Scientific Research and Technology Applications (SRTA), New Burg Al Arab, Alexandria 21934, Egypt; halaeladawi@hotmail.com; 6Department of Botany and Microbiology, College of Science, King Saud University, P.O. Box 2455, Riyadh 11451, Saudi Arabia; ssalmen@ksu.edu.sa; 7Department of Botany, Hindu College Moradabad (Mahatma Jyotiba Phule Rohilkhand University, Bareilly 244001, India; mjavedansari@gmail.com; 8Department of Agronomy, Institute of Food and Agricultural Sciences, University of Florida, Gainesville, FL 32611, USA; naeemkhan@ufl.edu; 9Department of Microbiology, PSGVP Mandal’s Arts, Science, and Commerce College, Shahada 425409, India; 10Asian PGPR Society for Sustainable Agriculture, Auburn University, Auburn, AL 36830, USA

**Keywords:** *Bacillus amyloliquefaciens* VB7, *B. velezensis*, whole genome sequencing, antifungal action, NRPS gene clusters, antiviral action, antifungal action

## Abstract

Chemical pesticides have an immense role in curbing the infection of plant viruses and soil-borne pathogens of high valued crops. However, the usage of chemical pesticides also contributes to the development of resistance among pathogens. Hence, attempts were made in this study to identify a suitable bacterial antagonist for managing viral and fungal pathogens infecting crop plants. Based on our earlier investigations, we identified *Bacillus amyloliquefaciens* VB7 as a potential antagonist for managing *Sclerotinia sclerotiorum* infecting carnation, tobacco streak virus infecting cotton and groundnut bud necrosis infecting tomato. Considering the multifaceted action of *B. amyloliquefaciens* VB7, attempts were made for whole-genome sequencing to assess the antiviral activity against tomato spotted wilt virus infecting chrysanthemum and antifungal action against *Fusarium oxysporum f.* sp*. cubense* (*Foc*). Genome annotation of the isolate *B. amyloliquefaciens* VB7 was confirmed as *B. velezensis* VB7 with accession number CP047587. Genome analysis revealed the presence of 9,231,928 reads with an average read length of 149 bp. Assembled genome had 1 contig, with a total length of 3,021,183 bp and an average G+C content of 46.79%. The protein-coding sequences (CDS) in the genome was 3090, transfer RNA (tRNA) genes were 85 with 29 ribosomal RNA (rRNA) genes and 21 repeat regions. The genome of *B. velezensis* VB7 had 506 hypothetical proteins and 2584 proteins with functional assignments. VB7 genome had the presence of flagellin protein FlaA with 987 nucleotides and translation elongation factor TU (Ef-Tu) with 1191 nucleotides. The identified ORFs were 3911 with 47.22% GC content. Non ribosomal pepide synthetase cluster (NRPS) gene clusters in the genome of VB7, coded for the anti-microbial peptides surfactin, butirosin A/butirosin B, fengycin, difficidin, bacillibactin, bacilysin, and mersacidin the Ripp lanthipeptide. Antiviral action of VB7 was confirmed by suppression of local lesion formation of TSWV in the local lesion host cowpea (Co-7). Moreover, combined application of *B. velezensis* VB7 with phyto-antiviral principles *M. Jalapa* and *H. cupanioides* increased shoot length, shoot diameter, number of flower buds per plant, flower diameter, and fresh weight of chrysanthemum. Further, screening for antifungal action of VB7 expressed antifungal action against *Foc* in vitro by producing VOC/NVOC compounds, including hexadecanoic acid, linoelaidic acid, octadecanoic acid, clindamycin, formic acid, succinamide, furanone, 4H-pyran, nonanol and oleic acid, contributing to the total suppression of *Foc* apart from the presence of NRPS gene clusters. Thus, our study confirmed the scope for exploring *B. velezensis* VB7 on a commercial scale to manage tomato spotted wilt virus, groundnut bud necrosis virus, tobacco streak virus, *S*. *sclerotiorum*, and *Foc* causing panama wilt of banana.

## 1. Introduction

Management of soil-borne pathogens in high-valued crops with chemical pesticides negatively impacts beneficial microflora dwelling in the rhizoplane. On the other hand, it also induces resistance among the population of soil-borne pathogens. The interest of scientists across the globe has been tweaked to devise an alternate management strategy. Amidst the various plant growth-promoting rhizobacteria, *Bacillus spp*. serve as a multifaceted bio-stimulant come immunomodulator with antifungal and antiviral action against several plant pathogens [1,2,3]. Moreover, strains of different *Bacillus spp*. inhibit soil-borne pathogens in varied environmental conditions [4,5]. 

To understand the treasure in the genome of *Bacillus spp.,* scientists have attempted to unravel the versatility of the genes. Genome mining of *Bacillus* species revealed the presence of 24 anti-microbial peptide genes (AMP) associated with the synthesis of non-ribosomal peptide synthetase gene clusters such as surfactin, mycosubtilin, subtilosin, subtilin, iturin, fengycin, mersacidin, bacilysin, ericin and bacillomycin [6]. Hitherto, the genetic makeup of *Bacillus* reveals the ability of antagonistic bacteria to produce anti-microbial compounds with volatile and nonvolatile nature, contributing towards the suppression of plant pathogens in a synergistic manner [7,8]. Similarly, *B. amyloliquefaciens* NJN-6 isolated from banana rhizoplane produced antifungal lipopeptides and diverse VOCs [9].

Considering the same attempts made to bioprime, the seeds with *B. amyloliquefaciens* VB7 coupled with polymer coating reduced the chilli damping-off caused by *Pythium ultimum* [9]. A liquid formulation of *B. amyloliquefaciens* VB7 suppressed the incidence of carnation stem rot, *Botrytis* leaf blight of lillium under protected condition [3,10]. Soil drenching and foliar application with a liquid formulation of *B. amyloliquefaciens* VB7 reduced the necrosis disease of cotton caused by *Tobacco streak virus* (TSV) and disease severity of GBNV infecting tomato under field conditions through the induction of defense-related genes [1,2,11]. Thus, considering the versatile nature of *B. amyloliquefaciens* VB7, attempts were made to mine the genome to explore the management of dreadful soil-borne pathogen *Fusarium oxysporum f. spp*. *cubense* (*Foc*) causing wilt in banana and tomato spotted wilt virus infecting chrysanthemum.

## 2. Materials and Methods

### 2.1. Comprehensive Genome Analysis and Genome Assembly

Genomic DNA of *Bacillus velezensis VB7* was isolated using the Quick-DNA Fungal/Bacterial kit (D6005) and subjected to whole-genome sequencing. The comprehensive genome analysis of antagonistic bacterial isolate VB7 was carried out by submitting the reads of VB7 isolate at PATRIC [12] and using TORMES-1.0 Unicyclker tools. Reads were processed using the TORMES-1.0 platform. Default parameters were used for quality trimming through trimmomatic (with the addition of adapter sequence removal of the P1 and A adapter sequences), after which an average read length of 149 bp remained. Assembly was performed using SPAdes (version 3.11.1), which produced contig length–3,021,183 bp, and a total assembly size of 3,021,183 bp. The mean GC content was 46.79%. All the bioinformatics settings were carried out by using the default settings. Trimmomatic was employed for quality trimming using the default parameters, with the addition of adapter sequence removal of the P1 and A adapter sequences. The genome of isolate VB7 was assembled using SPAdes [13]. Mapping quality-controlled reads to the contigs were carried out using BWA-MEM. 

### 2.2. Annotation of Genome Assembly

The genome of the isolate VB7 was annotated using RAST tool kit RASTtk [14] and genetic code 11 by comparing with other genomes in PATRIC to identify the genes of interest (speciality enes), a functional categorization (Subsystems), and to construct the phylogenetic tree (phylogenetic analysis). Proteins with functional assignments about different Enzyme Commission (EC) numbers were annotated as per the protocol by Schomburg et al. [15]. Proteins with Gene Ontology (GO) assignments were annotated as per the protocol described by Ashburner et al. [16], and proteins were mapped to KEGG pathways as per Kanehisa et al. [17]. Types of protein families in the genome of VB7 were annotated as per the protocol by Davis and his Coworkers [18]. Virulence genes were searched by screening the draft genome against virulence factors (VFDB, L, PATRIC-VF, and Victors DataBase) using Abricate and PARTIC. Any hit with coverage and identity below 90% was removed. Subsystem analysis of the genome was carried out to understand the involvement of a set of proteins that implement a specific biological process or structural complex [19], and annotation includes an analysis of the subsystems unique to each genome. Further, regions in the genome of *B. velezensis*-VB7 coding for secondary metabolites were identified using strictness ‘relaxed’ anti smash 5.1.0.

### 2.3. Comparative Genome Analysis 

Comprehensive characterization of different isolates related to the *B. velezensis*-VB7 strain, the assembled contigs of the test isolate VB7 and the selected 46 *Bacillus velezensis* genomes were analyzed using M1CR0B1AL1Z3R [20] for gene annotation, ortholog detection, sequence alignment and phylogeny reconstruction. The putative ORFs from each genome were extracted using Prodigal [21] in ‘normal’ mode, through an unsupervised machine learning approach to extract protein-coding ORFs. A homology search was conducted in which each ORF was queried against all other ORFs in the database. Markov Cluster (MCL) algorithm [22] with default parameters (inflation parameter = 2.0) was used to detect the high-confidence orthologous groups. 

Multiple sequence alignments (MSAs) were carried out for each identified orthologous group. Those sequences were first translated, and resulting protein sequences were then aligned using MAFFT, with the ‘auto’ flag, which automatically selected an appropriate MAFFT algorithm (L-INS-i, FFT- NS-i, or FFT-NS-2) according to the size of the analyzed dataset [23]. Sequences were then reverse-translated to compute codon-level alignments [24]. 

### 2.4. Phylogenetic Analysis

Mash/MinHash46 identified the closest reference and representative genomes. PATRIC global protein families (PGFams) were selected from these genomes to determine the phylogenetic placement of this genome [18]. The protein sequences from these families were aligned with MUSCLE [25], and the nucleotides for each of those sequences were mapped to the protein alignment. The joint set of amino acid and nucleotide alignments were concatenated into a data matrix, and RaxML [26] was used to analyze this matrix, with fast bootstrapping [27], which was used to generate the support values in the tree.

Reconstruction of the phylogenetic tree was carried out by a maximum-likelihood phylogenetic tree, which was carried out based on the concatenated protein MSA of all core genes, i.e., genes shared among all the compared strains using RAxML [26] with default parameters, the LG replacement matrix [28], and a discrete gamma distribution with four categories and an invariant category (LG+G+I) to account for among-site-rate variation. Subspecies clustering was done using a 79% dDDH threshold. Intergenomic distances were used to infer a balanced minimum evolution tree with branch support via FASTME 2.1.4 [29], including SPR post-processing. Branch support was inferred from 100 pseudo-bootstrap replicates each. The trees were rooted at the midpoint and visualized with PhyD3 [30].

### 2.5. Antiviral Efficacy of B. velezensis VB7 against TSWV

#### 2.5.1. Screening of Bacillus spp., against TSWV in Cowpea under Glasshouse Condition

Seven *Bacillus* strains viz., *Bacillus cereus* BSC 5 (JX036520), *B. pumilis*-BSC4 (JX036519), *B. amyloliquefaciens*-VB7 (KJ603234), *B. licheniformis*-B12 (KC540811), *B. subtilis* -BS TNAU1 (KC540800), *Pseudomonas fluorescence* Pf1 (AY818674) and *Ochrobactrum spp*. BSD 5 (JX036527) were assessed for the antiviral activity against TSWV at 10^8^ cfu/mL of the liquid suspension. Antiviral efficacy of *Bacillus spp*. in suppressing the symptom expression of TSWV was evaluated in vitro in cowpea variety Co-7. Forty-eight hours old cultures were inoculated into 100 mL nutrient broth and kept in a shaker at 120 rpm for 48 h. Broth containing 10^8^ cfu of respective bacterial strains were diluted to 0.5% and used for the assay. Inoculum of TSWV infecting chrysanthemum was maintained by inoculating onto the local lesion host (Cowpea-CO7). One gram of infected cowpea leaves with the local lesions were pooled and ground in prechilled sodium phosphate buffer pH-7.2 at 1 mL/g for further use [11]. Three methods were used to evaluate biocontrol agents’ efficiency in suppressing the symptom expression viz., post-inoculation spray (12 h interval), pre-inoculation spray (12 h interval), and simultaneous application using an atomizer immediately after mechanical inoculation of TSWV. Three replications were maintained with three plants in each pot with suitable untreated control. The total number of lesions per cm^2^ was recorded. Observations were pooled upon recording data from 15 blocks per replication. The best isolates were subjected to further evaluation in protected conditions. The parameters, such as percent incidence of TSWV, shoot length, the diameter of the shoot, number of flower buds, size of flower, and fresh weight of flower before harvesting, were recorded for evaluating the bio-efficacy of *Bacillus spp.* against TSWV.

#### 2.5.2. Antiviral Efficacy of B. velezensis VB7 and Phyto-Antiviral Principle against TSWV on Chrysanthemum (Mum Yellow) under Protected Cultivation

The field trial was conducted during 2020 at Masagal (11°31′6″ N, 76°54′25″ E) village of Nilgiri district, Tamil Nadu, India, under protected cultivation. The chrysanthemum variety Mum yellow was selected for the study. The experiment was replicated thrice per treatment with 12 m^2^ area/replication with a 10 /10 cm; simultaneously, untreated control was maintained. A bioefficacy test was carried out using three treatments: *B. amyloliquefaciens* VB7*, M. Jalapa*, and *H. cupanioides*. After transplanting, the same were delivered as a foliar spray at 15 days’ intervals 4 times. The bacterial suspensions and phyto-antiviral principles were delivered as a foliar spray either solely or in different combinations with phyto-antiviral principles. The experiment was laid out in randomized block design and replicated thrice. Treatments include T1-spraying with *B. amyloliquefaciens* VB7-0.5% at 10^8^ cfu/mL; T2-spraying with *M. Jalapa*–10%; T3-spraying with *H. cupanioides*-10%; T4-spraying with *B. amyloliquefaciens* VB7-0.5% at 10^8^ cfu/mL along with *M. Jalapa*-10%; T5-spraying with *B. amyloliquefaciens* VB7-0.5% at 10^8^ cfu/mL along with *H. cupanioides*-10%; T6-Combined spraying with *M. Jalapa* and *H. cupanioides* at 10%; T7-spraying with *B. amyloliquefaciens* VB7-0.5% at 10^8^ cfu/mL along with 10% *M. Jalapa* and 10% *H. cupanioides*;T8-untreated control.

### 2.6. Antifungal Activity of B. velezensis VB7 against Foc

The antifungal efficacy of bacterial endophytes was tested in vitro against *F. ox*-*ysporum f. sp*. *cubense* (Isolate-*Foc* KP-Acc. No. MW 436477) through the dual culture method. The mycelial disc (9 mm diameter) of seven day old culture of *Foc-*KP was placed at one side of the Petri plate containing PDA medium at 10 mm away from the periphery. *Bacillus spp*., (24 hold) were streaked onto the medium 10 mm away from the periphery, exactly opposite to the mycelial disc. The plates were incubated at 28 ± 2 °C for 7 days. The zone of inhibition against *Foc*-KP was measured after 7 days of incubation. The experiment was replicated thrice. Each replication consisted of 10 Petri plates per replication. The percent reduction of mycelial growth over untreated control was calculated using the formula:(1)$$\frac{C-T}{C}+\times 100$$

The experiment was repeated twice for confirmation [3], where C-Mycelial growth of the pathogen is the control, T-Mycelial growth of the pathogen is the dual plate technique.

### 2.7. Antifungal Activity of Volatile Organic Compounds (VOCs)/Nonvolatile Organic Compounds (VOCs) of B. velezensis VB7 against Foc KP 

The volatile and nonvolatile organic compounds were assessed for antifungal activity [31]. The VOC/NVOC compounds produced by *B. velezensis* VB7, diffused into PDA medium from the zone of inhibition, were excised using a sterile scalpel. Excised agar with the VOCs/NVOCs was mixed with HPLC grade acetonitrile in 1:4 ratios (5 g in 20 mL of HPLC grade acetonitrile). The mixture was sonicated twice for 30 s at 30% power of the sonicator for homogenization. After homogenization, samples were centrifuged and filtered to remove solid particles. The samples were dried in a vacuum flash evaporator (Rotrva Equitron Make). Subsequently, after removing the eluant, the final product was dissolved in 1 ml of HPLC grade methanol.

### 2.8. Characterization of VOCs/NVOCs Metabolites Produced during Trophic Interaction of Foc from the Zone of Inhibition of B. velezensis VB7 

The VOCs / NVOCs compounds secreted by *B. velezensis* VB7 during interaction with *Foc* produced the zone of inhibition. The biomolecules from the inhibition zone was extracted and characterized through GC/MS analysis. The difference in VOCs/NVOCs profile produced during the interaction of *B. velezensis* with *Foc* was compared with pathogen inoculated control, and *Bacillus velezensis* inoculated control through GC/MS (GC Clarus 500 Perkin Elmer Analysis) using the NIST version 2005 MS data library. 

### 2.9. Statistical Analysis

Mean differences of the treatment were evaluated with ANOVA by using Duncan’s multiple range test at 5% significance. All the data were statistically analyzed with IRRISTAT (version. 3/93, Biometrics unit, International Rice Research Institute) and interpreted.

## 3. Results

### 3.1. Comprehensive Genome Analysis

Comprehensive genome analysis revealed the presence of 9,231,928 reads with an average read length of 149 bp. Assembled genome had 1 contig, with a total length of 3,021,183 bp. The average G + C content was 46.79% (Appendix A). The shortest sequence of N50 length at 50% of the genome consisted of 3,021,183 bp. The L50 count, defined as the smallest number of contigs, whose length produced N50, was 1 (Appendix A). Analysis of mapping quality-controlled reads of the contigs with BWA-MEM confirmed an average genome coverage of 339X. Annotation statistics and comparing other genomes in PATRIC of the same species revealed a good genome quality. 

Genome annotation of the isolate VB7 using the RAST tool kit confirmed it as *Bacillus velezensis* VB7. The genome was placed under superkingdom and annotated with genetic code 11. Accordingly, the taxonomy of the genome pertained to a cellular organism > *Bacteria > Terrabacteria group > Firmicutes > Bacilli > Bacillales > Bacillaceae > Bacillus > Bacillus subtilis* group > *Bacillus amyloliquefaciens* group *> Bacillus velezensis*. The total number of protein-coding sequences (CDS) in the genome was 3090, transfer RNA (tRNA) genes were 85, along with 29 ribosomal RNA (rRNA) genes and 21 repeat regions (Appendix A). Further, the genome of *B. velezensis* VB7 had 506 hypothetical proteins and 2584 proteins with functional assignments. Proteins with functional assignments included 822 proteins with Enzyme Commission (EC) numbers, 684 with Gene Ontology (GO) assignments, and 607 proteins were mapped to KEGG pathways. The genome of VB7 was characterized with two types of protein families comprising of 2976 proteins pertaining to genus-specific protein families (PLFams) and 2976 proteins (PGFams) that belong to cross-genus protein families (Appendix A). Annotated genome of *B. velezensis* VB7 is displayed as a circular graphical representation, explaining the presence of contigs, CDS, RNA genes, coding sequences displaying the known anti-microbial resistance genes, virulence factors, GC content, and GC skew (Figure 1). 

Coding sequences of *B. velezensis* VB7 genome for MAMP genes revealed the presence of flagellin protein FlaA with 987 nucleotides spanning from 2,466,859 to 2,467,845 nt and translation elongation factor TU (Ef-Tu) with 1191 nucleotide (133,574 to 134,764 nt).

Multi-locus sequence typing (MLST) of *B. velezensis*-VB7 genome had, glpF (56), ilvD (~60), pta (~70), purH (~100), pycA (~90), rpoD (~70), tpiA (40) multi-locus sequences. Furthermore, the genome of *B. velezensis*-VB7 comprised specialty genes (Table 1), including virulence factor -3 (Source, Patric-VF), virulence factor -2 (Source, Victors), antibiotic resistance-4 (Source, CARD, NDARO), antibiotic resistance-34 (Source, PATRIC), transporter-164 (Source, CDB), drug target-42 (Source, Drug Bank) and drug target-1 (Source, TTD).

Annotation of the genome of VB7 using different databases for virulence factor features indicated the presence of genes coding for GTP-sensing transcriptional pleiotropic repressor, clpX coding for ATP-dependent Clp protease ATP-binding subunit, purB coding for adenylosuccinate lyase (EC 4.3.2.2) at SAICAR lyase (EC 4.3.2.2) responsible for virulence, RsfA coding for ribosomal silencing factor, purA coding for adenylosuccinate synthetase (EC 6.3.4.4) and flip coding for a flagellar biosynthetic protein (Appendix A). Subsystem analysis of the genome of VB7 reflected the presence of 1180 genes (1180-non-hypothetical; NIL-hypothetical) in the subsystem (37%) and 2045 genes (1539-non-hypothetical; 506-hypothetical), which were not grouped in subsystems, accounting for 63% (Figure 2). 

Annotation of the genome of *B. velezensis*-VB7 for the presence of subsystem revealed the occurrence of superclasses with different numbers of subsystem (SS) and families/genes (FG) are listed in Appendix A. Various superclasses present in the genome of VB7 are metabolism (73 SS/463 FG), cellular processes (27 SS/190 FG), stress response, defense, virulence (27 SS/116 FG), protein processing (36 SS/111 FG), energy (18 SS/104 FG), DNA processing (18 SS/104 FG), membrane transport (14 SS/63 FG), RNA processing (10 SS/36 FG), cell envelope (4 SS/15 FG), regulation and cell signaling (3 SS/11 FG) and miscellaneous (3 SS/7 FG).

### 3.2. Phylogenetic Analysis

Complete genome sequencing of the antagonistic bacterial isolate VB7 was identified as *B. velezensis*. The phylogenetic tree indicated the closeness between *B. velezensis* VB7 and *B. amyloliquefaciens* FZB 42 (Acc. No. 326423.5). The *B. amyloliquefaciens* DSM7 (Acc. No. 692420.6) and *B. siamensis* KCTC 13613 (Acc. No. 1177185.3) were clustered together. The isolate VB7 represent a separate clade and differed completely from *B.subtilis*, as shown in Figure 3. Reconstruction of the phylogenetic tree with concatenated protein MSA of all core genes shared among 46 strains of *B. velezensis* using RAxML and visualization of the tree using PhyD3 indicated the presence of variation between the strains of *B. velezensis* (Appendix A). Subspecies clustering at 79% dDDH threshold yielded seven species clusters. Moreover, strain VB7 was located in one of seven subspecies clusters. 

### 3.3. Comparative Genome Analysis 

Comparison of *B. velezensis* VB7 (CP047587) with the genome of other isolates of related species indicated a high variation in the number of encoded ORFs. The identified ORFs in the isolate VB7 is 3911, with a GC content of 47.22%. The genome with the lowest number of ORFs is *B. velezensis* (NCBI ACC.No. NZ-CP022556.1). It encodes 3653 ORFs. The genome with the highest number of ORFs is *B. velezensis* (NCBI ACC.No. NZ-CP017775.1) which encodes 4167 (Appendix A). Comparison of gene coding for unique proteins revealed the presence of 26 unique genes in the genome of VB7 in comparison with other *B. velezensis* genome strains bearing the accession numbers CP050462f, CP036527f, CP017747f and CP014990f. These unique genes were present in the (−) strand of the genome. Among the 26 genes, 12 were hypothetical proteins. Among the other 14 annotated unique proteins, seven proteins were present continuously in the negative strand. They are FMN reductase (NADPH) (EC 1.5.1.38), rod shape-determining protein Roda, uncharacterized N-acetyltransferase YedL, and cytochrome aa3-600 menaquinol oxidase subunit (I-IV) occupying the genomic position spanning from 2,733,044 to 2,740,283 nt continuously in the negative strand (Appendix A). Among the 684 annotated proteins, long-chain fatty acid-CoA ligase activity (GO:0004467) (0.9%), N-acetylmuramoyl-L-alanine amidase activity (GO:0008745) (0.9%), peptidoglycan glycosyltransferase activity (GO:0008955) (0.9%), DNA-directed DNA polymerase activity (GO:0003887) (1.2%) and protein histidine kinase activity (GO:0004673) (1.2%) were overrepresented.

### 3.4. Regions Coding for Anti-Microbial Peptides

Eight secondary metabolite regions were identified from the genome of VB7 (Table 2). Region 1 pertain to NRPS type member, similar to surfactin cluster with 82% similarity. It is located between the nucleotides spanning from 229,292–294,088 nt, accounting for a total length of 64,797 nt (Appendix A). Region 2, belonging to type member PKS-Like, was similar to cluster butirosin A/butirosin B spanning 6215,59–662,803 nt with 7% similarity (Appendix A). Region 3 was similar to type member terpene, spanning from 744,857–765,597 nt; however, it was not known to have similarity with known clusters (Appendix A). Region 4, spanning from 13, 24,708–1,432,535, was similar to the type of member NRPS transAT-PKS beta lactone, which resembled a fengycin cluster with 86% similarity (Appendix A). Region 5 of type member, trans AT-PKS-like, trans AT-PKS had 100% similarity with known cluster difficidin, spanning 1,545,898—1,652,076 nt (Appendix A). Region 6, corresponding to non-ribosomal peptide (NRP) cluster bacillibactin spanning from 2,089,753–2,141,546 nt belongs to type member NRPS bacteriocin (Appendix A). Similarly, region 7 related to another group was similar to cluster bacilysin spanning 2,673,492–2,714,910 nt (Appendix A). Further, region 8 belonging to type member lanthipeptide spanning from 2,856,858–2,880,046 nt resembled with 100% similarity to cluster mersacidin, the Ripp lanthipeptide (Appendix A).

### 3.5. Antiviral Efficacy of B. velezensis VB7 against TSWV Infecting Cowpea under Glasshouse Condition

Among the bacterial antagonists tested, the simultaneous application of *B. velezensis* VB7 and TSWV virus sap produced the mean lesion number of 1.41/cm^2^. It was followed by *B. licheniformis* (1.66 lesions/cm^2^). Comparative evaluation of the antiviral efficacy of phyto-antiviral principles revealed that simultaneous inoculation of TSWV along with 10% *M. Jalapa* effectively inhibited TSWV compared to *B. velezensis* VB7, which produced 0.75 lesions/cm^2^. It was followed by *H. cupanioides* (10%), contributing 0.83 lesions/cm^2^. However, 13.16 lesions/cm^2^ was observed in inoculated control. Moreover, none of the chemical inducers effectively suppressed the lesions produced by TSWV (Figure 4).

Pre-inoculation sprays with *M. Jalapa* and *H. cupanioides* at 10% effectively suppressed the infection of TSWV. It produced 1.16 and 1.33 mean lesions/cm^2,^ respectively. It was succeeded by *B. velezensis* VB7 and *B. licheniformis*, producing 2.50 and 2.75 lesions/cm^2^, respectively. However, 12.33 lesions/cm^2^ were observed in inoculated control. In post-inoculation spray with *M. Jalapa*, 1.91 lesions/cm^2^ were produced against 15.33 lesions/cm^2^ in inoculated control. It was subsequently followed by *H. cupanioides* (2.01 lesions/cm^2^) and *B. velezensis* VB7 (2.50 lesions/cm^2^). Comparison of different inoculation techniques indicated that post-inoculation spray with PGPR and phyto-antiviral principles was more ineffective than simultaneous and pre-inoculation spray. Among all the treatments, the root extract of *M. Jalapa* and seed extract of *H. cupanioides* followed by *B. velezensis* VB7 were effective in reducing the TSWV infection on cowpea, the local lesion host (Appendix A).

### 3.6. Evaluation of B. velezensis VB7 and Phyto-Antiviral Principle against TSWV on Chrysanthemum (cv. Mum Yellow) under Protected Cultivation

The antagonistic bacterial isolates and phyto-antiviral principles, including *B. velezensis* VB7, *M. Jalapa*, and *H. cupanioides* either alone or in combination, were sprayed once in 15 days the management of TSWV on chrysanthemum cultivar Mum yellow under protected conditions. Among all the treatments, combined application of *B. velezensis* VB7-0.5% (10^8^ cfu/mL), *M. Jalapa*-10% and *H. cupanioides*-10% had the highest mean shoot length of 86.40 cm, mean shoot diameter of 2.19 cm, the mean number of 13.20 flower buds per plant, mean flower diameter of 6.65 cm with a mean fresh weight of 5.60 g/flower. It was followed by the combined application of *M. Jalapa*-10% plus *H. cupanioides*-10%, which had the second-highest mean shoot length (81.90 cm), shoot diameter (2.11 cm), number of flower buds/plant (11.10), size of the flower (6.03 cm) and fresh weight of flower (5.13 g).

However, in untreated control, the mean shoot length, shoot diameter, number of flowers buds/plant, flower size and fresh weight of flower were 45.90 cm, 1.62 cm, 7.30 flower buds, 4.46 cm and 4.37 g, respectively. It revealed that combined application of *B. velezensis* VB7-0.5% (10^8^ cfu/mL) plus *M. Jalap*-10% and *H. cupanioides*-10% increased the growth and yield compared to untreated control (Appendix A). TAS-ELISA analyzed the titer of the virus. The ELISA results of the plants infected with TSWV revealed that OD value ranged from 0.411 to 0.474 in the plants sprayed with the combination of *B. velezensis* VB7-0.5% (10^8^ cfu/mL) plus *M. Jalapa*-10% and *H. cupanioides*-10% over untreated TSWV control plants with the OD value ranging from 0.786 to 0.975. (Appendix A). Furthermore, percent disease incidence was also reduced up to 4.83% in the plants sprayed with *B. velezensis* VB7-0.5% (10^8^ cfu/mL) plus *M. Jalapa*-10% and *H. cupanioides*-10% over untreated control plants (17.33%). These results confirmed that the combined application of *B. velezensis* VB7-0.5% (10^8^ cfu/mL) plus *M. Jalapa*-10% and *H. cupanioides*-10% increased the growth and flower yield by reducing the flower yield of the virus titer. 

### 3.7. In Vitro Antagonism of Bacterial Endophytes against Foc

Screening for the antagonistic activity of bacterial endophyte *B. velezensis* VB7 against *Foc* indicated that the mycelia growth of *Foc* was reduced up to 63.21% The mean percent inhibition of mycelial growth by other bacterial antagonists viz., *B. cereus* YEBN6, *B. endophyticus* YEBFL6, *B. siamensis* YEBN1, and *B. licheniformis* YEBFR6 ranged from 40 to 45% (Figure 5 and Appendix A)

### 3.8. Characterization of VOC/NVOC Biomolecules Produced by B. velezensis (VB7) from The Zone of Inhibition

VOCs/NVOCs compounds from the zone of inhibition produced during the ditrophic interaction of *B. velezensis* (VB7) with *Foc* revealed the presence of anti-microbial compounds, including hexadecanoic acid, linoelaidic acid, octadecanoic acid, clindamycin, formic acid, succinamide, furanone, 4H-pyran, nonanol and oleic acid (Appendix A). However, only seven biomolecules were identified in *B. velezensis* (VB7) in the PDA medium (Appendix A). The profile of the biomolecules produced from the zone of inhibition was different from the molecules produced by *B. velezensis* VB7 when cultured individually.

## 4. Discussion

The discovery of *B. velezensis* from brackish water of river Velez at Torredelmar in Malaga, southern Spain by Ruiz-Garcia et al. [32] during 2005 resulted in the subsequent identification of several strains of *B. velezensis* from different ecological regions across the globe. Several *B. amyloliquefaciens* are now reported as *B. velezensis*. Fan et al. [33], reported *B. velezensis* as a heterotypic synonym of *B. amyloliquefaciens. B. amyloliquefaciens* FZB42 was reported as type strain of *B. amyloliquefaciens* sub sp. *plantarum* [34]. It has been now reconfirmed as *B. velezensis*. The complete genome sequencing of the antagonistic bacterial isolate VB7 revealed the species identity as *B. velezensis* (CP047587). Based on the phylogenomic relationship, *B. velezensis* pertains to operational group *B. amyloliquefaciens* [33]. *B. velezensis* UFLA258 was compared with 115 strains. Among them, 19 strains deposited as *B. amyloliquefaciens* were reclassified as *B. velezensis* [35]. Comparison of the test isolate VB7 with the genome of other isolates and related species had a high variation in encoded ORFs. It might be due to the high rate of horizontal gene transfer or missing data in some genomes. Thus, it provides a gateway for further research concerning the source of differences in the number of ORFs among genomes. It was also interesting to note that Phage-like element PBSX protein xkdA (genomic position: 893,183–893,773) possesses LexA binding sites [36] in their putative promoter regions where LexA protein is involved in DNA repair and survival after DNA damage. Other unique proteins observed in the present study are cytochrome aa3–600 menaquinol oxidase proteins that play a vital role in energy conversion during vegetative growth [37]. Transcriptional regulator Xre, Cro/CI family, spore morphogenesis and germination protein YwcE, D-alanine-poly (phosphoribitol) ligase subunit 1 (EC 6.1.1.13) (involved in fatty acid biosynthesis), and Rod shape-determining protein RodA.

Further, many annotated genes have homology to known transporters [38], virulence factors [39,40], drug targets [41,42], and antibiotic resistance genes [43]. *Bacillus spp.* produce anti-microbial peptides responsible for antifungal, antibacterial, and antiviral action. Besides, secondary metabolites of *B. amyloliquefaciens*, *B. subtilis*, *B. velezensis* and *B. methylotrophicus,* serve as biosurfactants, siderophore, aid in nutrient mobilization and plant growth promotion [31,32,44,45,46,47,48,49,50]. Whole-genome sequence of *B. velezensis* isolate BZR 336g harbored gene with antifungal and plant growth-promoting properties. Several anti-microbial peptide genes, including surfactin, bacillomycin D, fengycin, bacillibactin, macrolactin, bacillaene and difficidin, were observed in the genome of BZR 336g [47]. Similarly, the draft genome sequence of our study isolate VB7 revealed the presence of anti-microbial peptide genes, such as surfactin, butrosin A/ butirosin B, NRPS transAT-PKS beta lactone, difficidin, bacillibactin, bacilysin, and mersacidin. Likewise, an earlier study by us using PCR confirmed the presence of NRPS gene clusters in *B. amyloliquefaciens* VB7, including ipa14 (Iturin), ituD, bamC (Bacillomycin), SFP (surfactin), bacA and bacD (Bacilysin), alba, and albF (subtilosin), and spaC and spas (subtilin) responsible for the antifungal action against carnation stem rot pathogen *Sclerotinia sclerotiorum* [3] which is now identified as *B. velezensis* VB7 based on whole-genome sequencing. Furthermore, metabolites of *B. amyloliquefaciens* VB7 also inhibited *S. sclerotiorum* by producing phenols and fatty acids. The presence of surfactin, iturin, bacillomycin, fengycin, mersacidin, bacilysin and subtilin in *Bacillus spp*., was also observed by several researchers [6,51].

Anti-microbial peptides such as bacilysin hindered glucosamine synthesis, fengycin altered cell wall permeability, bacillomycin hindered membrane permeability, and bacillibactin [52,53] chelated iron. Interestingly, complete genome sequencing of *B. velezensis* LS69 also revealed the presence of 10 NRPS gene clusters coding for the synthesis of macrolactin, bacillaene, difficidin, surfactin, fengycin, bacilysin, iturin A, amylolysin, and amylocyclicin. In addition, isolate LS69 had genes associated with enhanced plant growth and induction of innate immunity [54]. Our focus on the genome of *B. velezensis* VB7 revealed the presence of genes responsible for the synthesis of cyclic lipopeptides and polyketides. Further, they also triggered ISR to suppress the attack of plant pathogens [55]. Thus, the secondary metabolites of *Bacillus spp*. have either direct anti-microbial action against plant pathogens or inhibit the growth of competing microorganisms by depriving the needed iron essential for various activities [56]. A literature survey on the beneficial attributes of *B. velezensis* explained the potential role in the biological control of plant pathogens [2,3,11,34,48,57,58]. An investigation on the genome of *B. velezensis* VB7 also revealed the presence of various AMP genes associated with the biological control of plant pathogens.

Further, Zhang et al. [59] reported that bacillomycin secreted by *B. amyloliquefaciens* disrupted the plasma membrane of *Rhizoctonia solani*, and induced abnormalities in mycelia and conidia. NRPS gene cluster, Iturin penetrated cell membranes and induced cellular leakage [60]. Thus, inhibition of *Foc* in the present study by *B. velezensis* VB7 might be due to the presence of diverse NRPS-AMP genes present in the genome of the antagonist.

The antiviral activity based on the presence of different MAMP genes and AMP genes in the genome of *B. velezensis* VB7 was assessed. In our study, the antagonist bacteria *B.velezensis* VB7 and phyto-antiviral principles *M. Jalapa* and *H. cupanioides* effectively suppressed the TSWV in chrysanthemum plants (cv. Mum yellow) under protected cultivation. The maximum reduction in the incidence of TSWV was observed in the chrysanthemum cv. Mum yellow sprayed with the mixture of *B.velezensis* VB7*, M. Jalapa*, and *H. cupanioides.* Virus titter was several folds reduced compared to untreated control. Moreover, it also increased growth and yield parameters. Plant growth-promoting *Bacillus spp*, produced extracellular metabolites such as gibberellins, cytokinins, and indole acetic acid, which triggered the growth [49,61]. Further, they also reported that rhizosphere colonization and secretion of secondary metabolites by beneficial bacteria in the rhizosphere resulted in the alteration of root architecture by increasing root hair production and lateral roots that, in turn, increased water and nutrient uptake. In our investigation, plant growth promotion through the combined application of *B. velezensis* VB7 and *M. Jalapa* and *H. cupanioides* might have increased the uptake efficiency of the plants to absorb nutrition and would have favored the plant growth and reduced virus titter compared to the untreated control.

Earlier works by various researchers also indicated that *Bacillus* species quenched viral infection in plants. Delivering consortia of *Bacillus* sp., *Pseudomonas* sp. and *Burkholderia* sp. reduced cotton leaf curl incidence [62]. In our study, the virus titter was drastically reduced to 0.474 in infected plants treated with *B. velezensis* VB7 (0.5% at 10^8^ cfu/mL) plus *M. Jalapa* (10%) and *H. cupanioides* (10%) compared to infected plants in the untreated control (0.975). Similar reports on the reduction of virus titter upon application of *Bacillus spp*., has been observed for banana bunchy top virus [63], cucumber mosaic virus [64] and tomato mosaic virus [65]. Apart from antiviral action, *Bacillus spp*., also promoted plant growth. The endogenous pool of phytohormones was increased in the presence of rhizobacteria [66]. Islam et al. [67] reported that *Bacillus spp*., increased germination percentage, seedling vigo and improved nitrogen content in cucumber plants. Secretion of growth hormone cytokinin by *B. megaterium* contributed to growth promotion. Subsequently, cytokinin receptors CrE1, AHK2, and AHK3 genes promoted plant growth. In our study, the combined application of *B. velezensis* VB7 with phyto-antiviral principles has reduced the virus titer and increased chrysanthemum’s yield and plant growth. Earlier work also emphasizes that MAMP genes such as flagellin (Flag) and elongation factor (EF-Tu) induce plant defense in tomatoes against GBNV. Drenching the clones of *Agrobacterium tumefaciens* EHA105 expressing the MAMP genes, flagellin (Ag-Ba. Flag), and elongation factor-TU (Ag-Ba. EF-Tu) in root zone induced both SAR and ISR against GBNV in tomato due to the increased transcript levels of NPR1, PR1, MAPKK1 and WRKY33BB [2]. Likewise, the challenging of chilli plants with *B. amyloliquefaciens* CRN9 suppressed the infection of GBNV in chilli due to the induction of genes NPR1, PAL, PO, SAR8.2, PDF, LOX, EAS1, and WRKY33 associated with ISR/SAR pathway [1]. Likewise, analysis for the presence of MAMP genes in the genome of *B. velezensis* VB7 (*B. amyloliquefaciens* VB7) revealed the presence of flagellin protein FlaA with 987 nucleotides spanning from 2,466,859–2,467,845 nt and translation elongation factor TU (Ef-Tu) with 1191 nucleotide (133,574–134,764 nt). Thus, these MAMP genes might have played a vital role in MAMP triggered immunity against TSWV infecting chrysanthemum.

Not only were the plant viruses suppressed, but also the soil-borne pathogen *F. oxysporum* f.sp. *cubense* causing panama wilt of banana was also suppressed by *B. velezenzis* VB7. It inhibited the mycelial growth of *Foc* by more than 60% over control. *B. velezenzis* CT32 inhibited the mycelial growth of *Verticillium dahliae* and *F. oxysporum*, causal agents of strawberry vascular wilt up to 66.67% [50]. Hong et al. [68] tested the antifungal activity of *B.amyloliquefaciens* against green and blue mold and sour rot caused by *Penicillium digitatum, P. italicum* and Ge*otrichum citri-aurantii* in mandarin fruit. *B. amyloliquefaciens* isolated from the banana plants exposed to maximum disease pressure by *Foc* promoted banana growth and antifungal action by producing anti-microbial peptides and volatile organic compounds [44]. *B. amyloliquefaciens* NJN-6 W19 produced diverse antifungal metabolites, including lipopeptides VOCs [44]. In the present study, the biomolecules produced by *B. velezenzis* VB7 were identified as hexadecanoic acid, linoelaidic acid, octadecanoic acid, clindamycin, formic acid, succinamide, furanone, 4H-pyran, nonanol, and oleic acid. Antifungal activity of formic acid against *F. oxysporum* was reported by Hasan et al. [69]. Liu et al. [70] reported the inhibitory effect of oleic acid against *F. oxysporum* and growth promotion in tomato and cucumber seedlings. Pohl et al. [71] confirmed the antifungal activity of nonanoic acid, oleic acid, hexadecanoic acid, and tridecanoic acid against various filamentous plant parasitic fungi. Broad-spectrum anti-microbial activity of clindamycin was reported by Guay et al. [72]. Walters et al. [73] reported the antifungal activity of linoleic acid against *Rhizoctonia solani, Pythium ultimum*, and *P. avenae* at 1000 μM. Subsystem analysis of the whole genome of VB7 in our investigation confirmed the presence of 73 subsystems coding for 463 functional genes associated with cellular metabolism. These metabolic genes might be responsible for synthesizing diverse VOCs /NVOCs compounds responsible for antifungal action. Thus, MAMP genes, NRPS gene clusters and metabolic genes in *B. velezensis* VB7 (CP047587) might have contributed to the antiviral and antifungal activity. Hence, antagonistic *B. velezensis* VB7 has opened new vistas with a broad spectrum of action for both viral and fungal diseases in crop plants.

## 5. Conclusions

Our earlier investigation on bacterial antagonists emanated in identifying *B*. *amyloliquefaciens* VB7 with a broad spectrum of action against *S. sclerotiorum* causing stem rot of carnation, tobacco streak virus infecting cotton, and groundnut bud necrosis infecting tomato. The broad-spectrum action, genome annotation of *B. amyloliquefaciens* VB7 revealed the identity as *B. velezensis* VB7 (CP047587). Assembled genome facilitated to detect the presence of NRPS gene clusters encoding surfactin, butirosin A/butirosin B, fengycin, difficidin, bacillibactin, bacilysin and mersacidin the Ripp lanthipeptide responsible for antifungal and antiviral action through wet lab confirmation. Thus, the present investigation has opened new vistas for exploring *B. velezensis* VB7 on a commercial scale to manage tomato spotted wilt virus, groundnut bud necrosis virus, tobacco streak virus, *S*. *sclerotiorum*, and *Foc* causing panama wilt of banana.

## Figures and Tables

**Figure 1 microorganisms-09-02511-f001:**
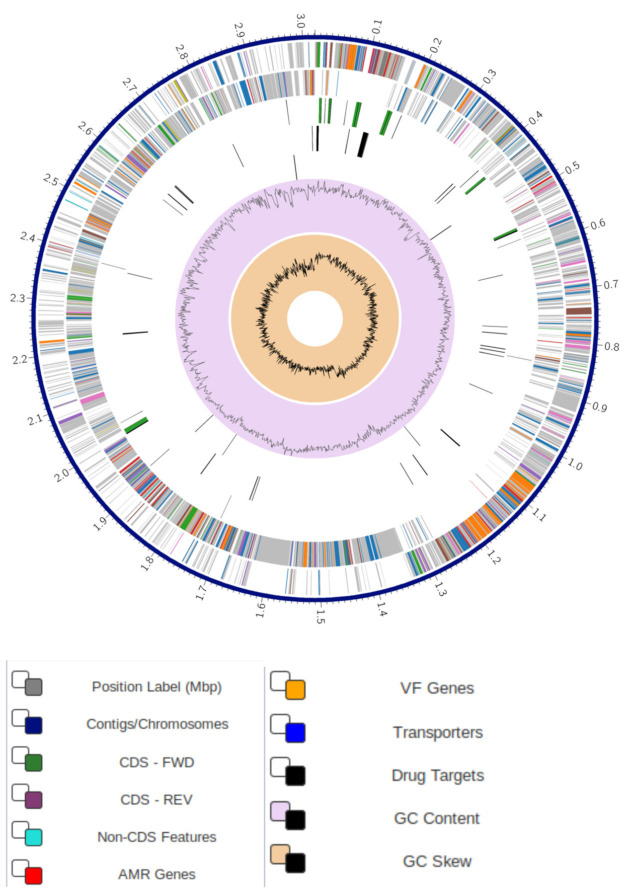
Circular Graphical Assembly of the genome of *Bacillus velezensis* isolate VB7. Note: The outer to inner rings, the contigs, CDS on the forward strand, CDS on the reverse strand, RNA genes, CDS with homology to known anti-microbial resistance genes, CDS with homology to known virulence factors, GC content, and GC skew. The colors of the CDS on the forward and reverse strand indicate the subsystem that these genes belong to (see Subsystems below).

**Figure 2 microorganisms-09-02511-f002:**
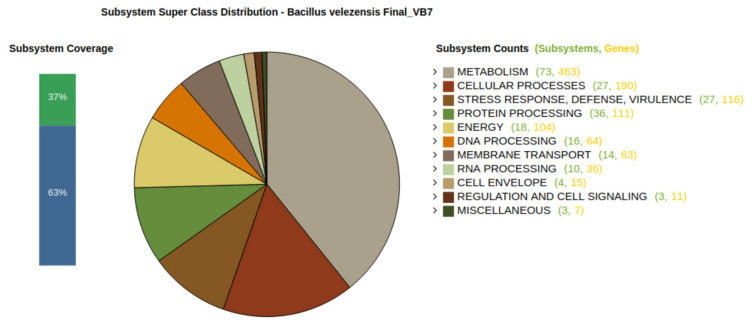
Subsystem Analysis of the Genome *Bacillus velezensis*-VB7.

**Figure 3 microorganisms-09-02511-f003:**
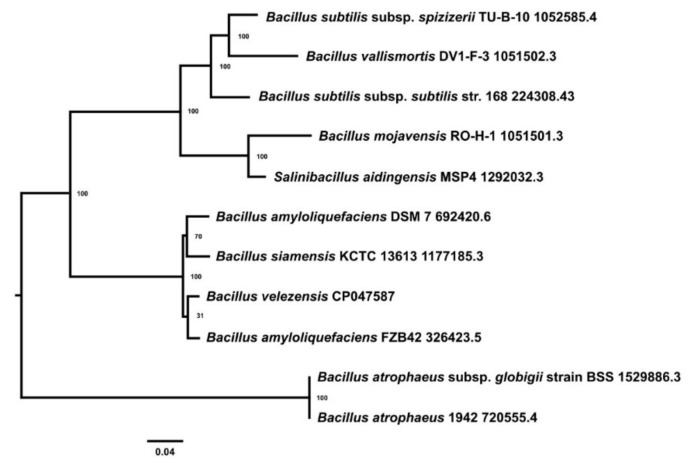
Phylogenetic Analysis of the Genome of *Bacillus velezensis*-VB7 CP047587.

**Figure 4 microorganisms-09-02511-f004:**
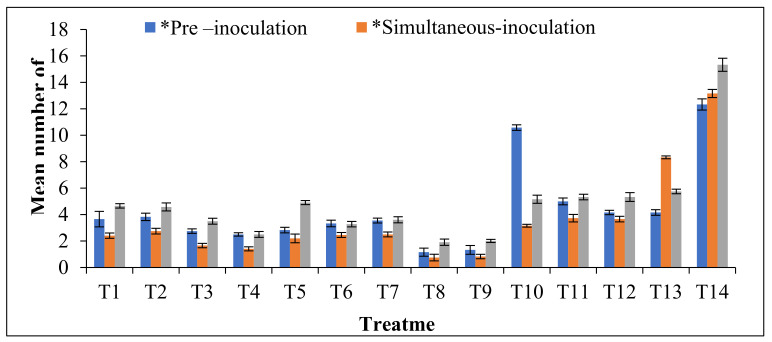
In vitro screening for the antiviral activity of bio-control agents, botanicals, and defense inducers against TSWV on Cowpea (var. Co 7).Note:T1-*Bacillus cereus* BSC5-0.5% at 10^8^ cfu/mL; T2-*B. pumilis* BSC4 -0.5% at 10^8^ cfu/mL; T3-*B. licheniformis* B12-0.5% at 10^8^ cfu/mL; T4-*B. velezensis* VB7-0.5% at 10^8^ cfu/mL; T5-*B. subtilis* BAG3-0.5% at 10^8^ cfu/mL; T6-*Pseudomonas fluorescens* Pf1-0.5% at 10^8^ cfu/mL; T7-*Ochrobactrum spp*. BSD 5-0.5% at 10^8^ cfu/mL; T8-*Mirabilis jalapa* (MJ)-10%; T9-*Harpulia cupan*ioides (HC) 10%; T10-Potassium sulphate (K_2_SO_4_)-0.1%; T11-Potassium silicate (K_2_SiO_3_)-0.1%; T12-Potassium thiosulphate (K_2_S_2_O_3_)-0.1%; T13-Chitosan-1%; T14-Control.

**Figure 5 microorganisms-09-02511-f005:**
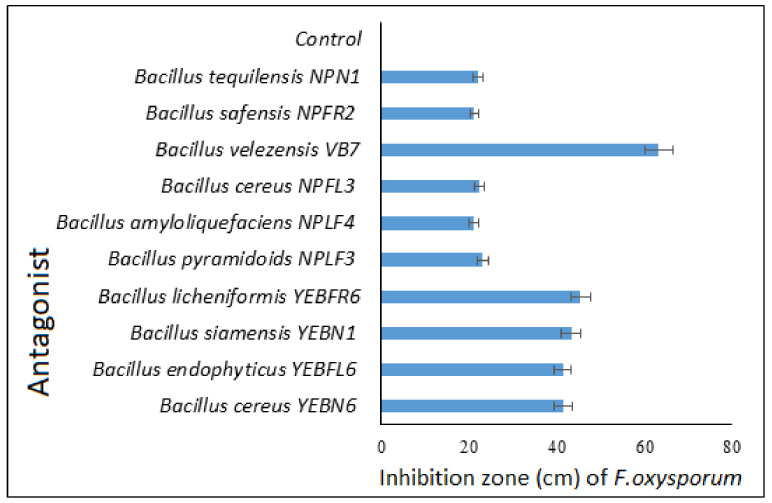
Antifungal activity of bacterial antagonists against *F. oxysporum* f.*sp*. *cubense in* vitro.

**Table 1 microorganisms-09-02511-t001:** Specialty genes in the genome of *B. velezensis*-VB7.

Specialty Genes	Source DataBase	Gene Numbers
Virulence Factor	PATRIC_VF	3
Virulence Factor	Victors	2
Antibiotic Resistance	CARD, NDARO	4
Antibiotic Resistance	PATRIC	34
Transporter	TCDB	164
Drug Target	Drug Bank	42

**Table 2 microorganisms-09-02511-t002:** Regions Coding for Secondary Metabolites in the Genome of *Bacillus velezensis*-VB7.

Region	Type	Nucleotide		Most Similar Known Cluster		Similarity %
		From	To			
Region 1	NRPS	229,292	294,088	Surfactin	NRP lipopeptide	82
Region 2	PKS like	621,559	662,803	Butirocin A/ButirocinB	Saccharide	7
Region 3	Terpene	744,857	765,597	-	-	-
Region 4	NRPS-Trans At- PKS beta lactone	1,324,708	1,432,535	Fengycin	NRP	86
Region 5	Trans AT-PKS-like trans At-PKS	1,545,989	1,652,076	Difficidin	Polyketide + NRP	100
Region 6	NRPS bacteriocin	2,089,753	2,141,546	Bacillibactin	NRP	100
Region 7	Other	2,673,492	2,714,910	Bacilysin	Other	100
Region 8	Lanthipeptide	2,856,858	2,880,046	Mersacidin	Ripp-Lanthipeptide	100

## Data Availability

All the data is available in the manuscript and its Appendix A.

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
