# Peer review of "Mining the Genome of Bacillus velezensis VB7 (CP047587) for MAMP Genes and Non-Ribosomal Peptide Synthetase Gene Clusters Conferring Antiviral and Antifungal Activity"

_microorganisms, 2021, doi:10.3390/microorganisms9122511_

Round 1

Reviewer 1 Report

Manuscript Title: Mining the genome of Bacillus velezensis VB7 for MAMP genes and Non-Ribosomal Peptide Synthetase gene clusters conferring antiviral and antifungal activity

Manuscript ID: microorganisms-1445617

The manuscript written by Saravanan et al. reports some interesting results. In this work, the authors have investigated the the genome of Bacillus velezensis VB7 for MAMP genes and Non-Ribosomal Peptide Synthetase gene clusters conferring antiviral and antifungal activity. Moreover, the authors screened antifungal action of VB7 expressed antifungal action against Foc in vitro by producing VOC/NVOC compounds. Further research on this area may result in the development of a new approach for control of tomato spotted wilt virus, groundnut bud necrosis virus, tobacco streak virus, S. sclerotiorum, and Foc causing panama wilt of banana.

In general, this paper is clearly laid out, well planed and easy to read. The experiments are well designed and appropriate controls are presented. Some specific suggestions or questions are listed below:

  1. Please use full name for NRPSfor the first time in abstract.
  2. Introductionis too long. Better organize the subject and just describe the main results.
  3. Introduction is easy to read but needs a little completed. For example, in the introduction, there is a characterization of the main scientific gaps, but there is no clear connection with the objectives of the article. I suggest the authors make this connection.
  4. Table 1. Assembly details of the antagonistic bacterial isolate B. velezensis VB7 - CP047587. This tablecan be put as Supplementary Materials.
  5. Figure 3. Phylogenetic analysis of the genome of Bacillus velezensis – VB7 CP047587. This figure should be improved. The figure is not clear, and the genus names should be in italic.
  6. Figure 4. Violin plot representing the dispersion of the identified ORFs count per genome. The figure can be put as Supplementary Materials.
  7. Discussion: why the author did not compare the antifungal activityof Bacillus velezensis VB7 with that of other Bacillus strains based on the literature such as doi: 10.1016/j.postharvbio.2013.10.004. Bacillus strains are known for their metabolic capability and environmental versatility as well as for their ability to manage bacterial and fungal pathogens infecting crop plants. Authors should add more information into this section and cite the recent research into the field.
  8. Conclusion section should be revised and what research gap fulfilled by this research and possibility for future after this. Please elaborate the conclusion.
  9. References: Many of the references have been superceded and more modern ones are required, such as Lemma, E., 1995;  Keerthana, M, 2000;  Saier Jr, 2002.

Reviewer 1 Report

Reviewer 1 has indicated the corrections to be made in the manuscript via, track change. Hence, all the corrections mentioned via track changes has been carried out and the same has been included. The details of the corrections made are appended in the table1.

S.

No

Line number

Comments to Authors

Author’s response

1

97

the (remove italic for the) Tobacco streak virus

Correction has been carried out as suggested.

2

129

insert space after.

Correction has been carried out as suggested.

3

150 

All sequences What?

Thanks for the suggestion given by the reviewer for further improving the quality of the manuscript. As per the suggestion, the statement has been changed.

4

187

How to inoculate TSWV on cowpea?

Thanks for the reviewer comment for improving the clarity of the manuscript. As per the suggestion by reviewer mechanical inoculation of TSWV on cowpea has been included. This is for your kind consideration.

5

221

need citation

Citation has been added for calculation of per cent reduction over control, as suggested by the reviewer.

6

Pg No 9

Line 35

Amon?

Thanks for the reviewer for his suggestion. Amon has been changed as Among

7

Pg No 12

Line 81

Figure 5

1. y axis, mean number of? what?

2. indicates gray bar too

Thanks for the reviewer for his suggestion. Correction has been carried out as suggested.

8

Pg 12

Line 99

Supplementary Figure S10

Thanks for the reviewer suggestion. Supplementary files has been included

9

Line 134

Fig 6. How about statistical analysis among antagonist?

Thanks for the reviewer suggestion. Statistical analysis for antagonistic action has been included in statistical analysis under section 2.9 of material and methods.

10

IPg 13

Line 150 

spp.?

Thanks for the reviewer suggestion. Correction has been carried out as suggested. The statement has been rephrased. 

11

Pg 14

Line 173

spp.?

Thanks for the reviewer suggestion. Correction has been carried out as suggested. The statement has been rephrased 

12

Pg 15
Line 224

B. velezensis VB7 and (non italic) M. Jalapa

Thanks for the reviewer suggestion. Correction has been carried out as suggested.

13

Pg 15

Line 256

F. oxysporum f.spp. cubense?

Thanks for the reviewer suggestion. Correction has been carried out as suggested. F. o.  f.spp. cubense changed as F. oxysporum f.spp. cubense

Reviewer 2 Report

Some minor points need to be addressed, please find an attachment.

Reviewer 2 Report

In general, this paper is clearly laid out, well planned and easy to read. The experiments are well designed and appropriate controls are presented. Some specific suggestions or questions are listed below:

Comments to authors

 Please use full name for NRPS for the first time in the abstract.

Authors’ response: Correction has been carried out as suggested. Line 42

  1. Introduction is too long. Better organize the subject and just describe the main results.

Authors’ response: Thanks for the suggestion given by the reviewer to improve the manuscript clarity. The introduction part has been rephrased.

  1. The introduction is easy to read but needs a little completed. For example, in the introduction, there is a characterization of the main scientific gaps, but there is no clear connection with the objectives of the article. I suggest the authors make this connection.

Authors’ response: Correction has been carried out as suggested by the reviewer. The introduction chapter has been rephrased. Line No. 59-106

  1. Table 1. Assembly details of the antagonistic bacterial isolate B. velezensis VB7 - CP047587. This table can be put as Supplementary Materials.

Authors’ response: Correction has been carried out as suggested by the reviewer. Table 1 has been moved to supplementary information’s. Line No. 260

  1. Figure 3. Phylogenetic analysis of the genome of Bacillus velezensis – VB7 CP047587. This figure should be improved. The figure is not clear, and the genus names should be in italic.

Authors’ response: Correction has been carried out as suggested by the reviewer. Figure 3 has been recreated with improved clarity. The scientific name of bacteria has been indicated in italic. Line 46

  1. Figure 4. Violin plot representing the dispersion of the identified ORFs count per genome. The figure can be put as Supplementary Materials.

Authors’ response: Correction has been carried out as suggested by the reviewer. Figure 4 has been moved to supplementary information. Line 48

  1. Discussion: why the author did not compare the antifungal activity of Bacillus velezensis VB7 with that of other Bacillus strains based on the literature such as doi: 10.1016/j.postharvbio.2013.10.004. Bacillus strains are known for their metabolic capability and environmental versatility as well as for their ability to manage bacterial and fungal pathogens infecting crop plants. Authors should add more information into this section and cite the recent research into the field.

Authors’ response: Correction has been carried out as suggested by the reviewer. As the reviewer suggested doi 10.1016/j.postharvbio.2013.10.004 has been cited.

  1. Conclusion section should be revised and what research gap fulfilled by this research and possibility for future after this. Please elaborate the conclusion.

Authors’ response : Correction has been carried out and the conclusion has been rephrased as suggested by the reviewer. Line 283-302 

  1. References: Many of the references have been superseded and more modern ones are required, such as Lemma, E., 1995; Keerthana, M, 2000;  Saier Jr, 2002.

Authors’ response: Correction has been carried out as suggested by the reviewer.

Round 2

Reviewer 1 Report

The authors has improved the manuscript.